# Synthesis of 4,5-Dihydro-1*H*-[1,2]dithiolo[3,4-*c*]quinoline-1-thione Derivatives and Their Application as Protein Kinase Inhibitors

**DOI:** 10.3390/molecules27134033

**Published:** 2022-06-23

**Authors:** Svetlana M. Medvedeva, Khidmet S. Shikhaliev

**Affiliations:** Department of Organic Chemistry, Faculty of Chemistry, Voronezh State University, 1 Universitetskaya Sq., 394018 Voronezh, Russia; chocd261@chem.vsu.ru

**Keywords:** 1,2-dithiol-3-thione, 2,2-disubstituted-1,2-dihydroquinolines, dithioloquinolinethiones, non-selective anti-kinase activity, bioactivity profiles prediction in silico, PASS online

## Abstract

This study represents the design and synthesis of a new set of hybrid and chimeric derivatives of 4,5-dihydro-4,4-dimethyl-1*H*-[1,2]dithiolo[3,4-*c*]quinoline-1-thiones, the structure of which the tricyclic fragment linearly bound or/and condensed with another heterocyclic fragment. Using the PASS Online software, among the previously synthesized and new derivatives of 1,2-dithiolo[3,4-*c*]quinoline-1-thione we identified 12 substances with pleiotropic activity, including chemoprotective and antitumor activity. All the synthesized derivatives were screened for their inhibitory assessment against a number of kinases. Compounds which exhibited prominent inhibition percentage in cells (>85%) were also examined for their inhibitory efficiency on human kinases via ELISA utilizing sorafenib as a reference standard to estimate their IC_50_ values. It was revealed that compounds **2a**, **2b**, **2c**, and **2q** displayed a significant inhibition JAK3 (IC_50_ = 0.36 μM, 0.38 μM, 0.41 μM, and 0.46 μM, respectively); moreover, compounds **2a** and **2b** displayed excellent activities against NPM1-ALK (IC_50_ = 0.54 μM, 0.25 μM, respectively), against cRAF[Y340D][Y341D], compound **2c** showed excellent activity, and compound **2q** showed weak activity (IC_50_ = 0.78 μM, 5.34 μM, respectively) (sorafenib IC_50_ = 0.78 μM, 0.43 μM, 1.95 μM, respectively). Thus, new promising preferred structures for the creation of drugs for the treatment of cancer and other multifactorial diseases in the future have been found.

## 1. Introduction

Interest in 3*H*-1,2-dithiol-3-thione (DTT) derivatives has developed continuously but unevenly for nearly 140 years. For the first time, a compound containing a 1,2-dithiol-3-thione ring (4,5-dimethyl-1,2-dithiol-3-thione **I**, Figure 1) was obtained in 1884 by Barbaglia [1]. The study of the chemistry of these five-membered pseudoaromatic compounds started to develop later, in the middle of the 20th century, after the appearance of a number of studies devoted to the synthesis of DTT derivatives and their isolation from cruciferous plants [2,3,4,5,6,7].

In the mid-1980s, there was a surge in interest in DTT derivatives after the introduction of Oltipraz (4-methyl-5-(2-pyrazinyl)-1,2-dithiol-3-thione **II**, Figure 1) into clinical practice as an antiparasitic agent against Schistosoma mansoni (1985, pharmaceutical company Aventis) [8]. Further studies have shown that oltipraz also exhibits cancer-preventive activity and can be used in the treatment of cancer [9,10,11,12]. Oltipraz and its analogues exhibit a cancer-preventive effect via the induction of cell cytoprotective enzymes involved in the detoxification of carcinogens and via stimulation of the restoration of DNA damaged by carcinogens [13,14], or by the activation of transcription factors [15,16].

3*H*-1,2-Dithiol-3-thiones inhibit the action of the resulting hydrogen peroxide, protect mitochondria from oxidative stress, and increase their antioxidant potential [17,18]. The 1,2-dithiol-3-thione fragment is an effective endogenous donor of hydrogen sulfide, an important gaseous signaling molecule (gas transmitter) in the human body, and is involved in the regulation of a number of physiological and pathophysiological processes [19,20]. One of the worldwide most-studied hydrogen sulfide donors is 5-(4-hydroxyphenyl)-3*H*-1,2-dithiole-3-thione ((ADT-OH) **III**, Figure 1), which inhibits apoptosis and removes ROS through stimulation of glutathione and glutathione-*S*-transferase [20].

DTT, as H_2_S-releasing compounds, not only exhibit anticancer properties, but also analgesic activity [21], and protect cardiomyocytes from ischemic cell death [22]. Hybrids of DTT and non-steroidal anti-inflammatory drugs (aspirin, diclofenac) are being developed as promising drugs for the treatment of inflammatory diseases of various etiologies [23,24]. The presence of fragments with different pharmacotherapeutic profiles in one molecule can be useful for a reduction in side effects, enhancement of the action of a drug or reduction in resistance to it, and also for expanding the range of its application [25,26].

Therefore, the design of hybrid or chimeric compounds in the structure of which the dithiol ring is combined with another heterocyclic fragment (for example, azaheterocycle) with a certain pharmacological activity is of current interest. Among azaheterocycles, 2,2-disubstituted-1,2-dihydroquinolines functionalized at the benzene ring and/or the nitrogen atom (DHQ), which have a wide spectrum of biological activity—anticoagulant, antimalarial, antiparasitic, antibacterial, antidiabetic, anti-inflammatory, neuroprotective, and hepatoprotective activity—are of particular interest [27,28,29,30,31,32,33].

For some representatives of hybrid 4,5-dihydro-4,4-dimethyl-1*H*-[1,2]dithiolo[3,4-*c*]quinoline-1-thiones (DTT-DHQ), in which dithiolthione and hydroquinoline cycles are annelated via the [*c*] bond of the latter (Figure 2), we have previously revealed an inhibitory effect on the blood coagulation system [34]. We also studied the antimicrobial antifungal activity of these polycyclic dithioloquinolinethiones and identified compounds with an activity exceeding the effects of ampicillin and streptomycin. Their antifungal activity was higher than that of the reference drugs ketoconazole and bifonazole [35]. Recently, using the PASS Online (Prediction of Activity Spectra for Substances) software [36], we determined the probable anti-inflammatory effect of various dithioloquinolinethione derivatives and experimentally confirmed that the anti-inflammatory activity of the studied compounds is comparable to or higher than that of the reference drug indomethacin [37] (Figure 2).

Due to the discovered multiple (pleiotropic) action of annelated DTT, the search for new types of activity, in particular, anti-cancer activity, in previously obtained dithioloquinolinethione derivatives, the targeted synthesis and study of the activity of new representatives of this series are of considerable interest for medicinal chemistry and pharmaceutical science.

It is known that the causes of cancer, diabetes, inflammatory processes, and other multifactorial diseases are mutation processes and/or activation of enzymes of the protein kinase family [38,39]. The human genome contains 518 kinases that transfer the γ-phosphate of ATP to the hydroxyl group of tyrosine, serine, or threonine residues of the substrate. Protein kinases play a key role in cell proliferation, metabolism, and apoptosis; therefore, they have become a target for anticancer drug therapy [40]. One of the strategies of antitumor therapy is the inhibition of protein kinases by low-molecular-weight ATP mimetic compounds which block oncogene-induced cell signaling pathways, affect the proteins that regulate gene functions, induce the apoptosis of cancer cells, etc. [41].

Most of the early kinase inhibitors exhibited poor selectivity for undesirable targets such as ion channels, cytochrome P450 (CYP), and other proteins, causing side effects [42,43]. In order to minimize the risk of side effects, a current trend is the development of targeted (highly selective) kinase inhibitors with directed pathogenetic action [44]. However, a relatively small number of selective kinase inhibitors have been approved in practice, since the clinical use of these inhibitors has led to the emergence of drug-resistant tumors. In many patients, the response to small molecule kinase inhibitors was accompanied by tumor recurrence, making these inhibitors less effective than expected [45,46,47,48]. This resistance was associated with a number of mechanisms that include the gene amplification of oncogenic kinase [49] and alternative signaling pathways or signaling plasticity [50].

The insensitivity of the drug to kinase mutations is the main task of molecular design and the synthesis of targeted antitumor organic compounds. Changing the selectivity profile in the treatment of complex diseases such as cancer can lead to an improvement in the therapeutic quality of the compound. In the last decade, studies and clinical practice for drugs have established the advantages of multi-inhibition in cancer therapy in comparison with monotarget inhibition. In antitumor therapy, multikinase inhibition may be useful primarily for the inhibition of rapidly mutating kinases [51,52,53,54,55,56]. Multitarget drug discovery (MTDDa) drugs that affect more than one target link can provide super-efficacy and safety comparable to monotargeted drugs [57,58].

The aim of this study was molecular modeling, computer screening for potential antitumor activity, synthesis of linearly bound (hybrid molecules) and condensed derivatives (chimeric molecules) of 4,5-dihydro-1H-[1,2]dithiolo[3,4-*c*]quinoline-1-thiones, and the investigation of their inhibitory activity against a number of protein kinases, namely, NPM1-ALK, ALK, EGFR[L858R][T790], cRAF[Y340D][Y341D], JAK2, and JAK3.

Using the PASS Online software, among the previously synthesized and new hybrid and chimeric derivatives of 1,2-dithiolo[3,4-c]quinoline-1-thione we identified 12 substances with pleiotropic activity, including chemoprotective and antitumor activity, and experimentally confirmed their inhibitory activity. For the leading compounds (phenylpiperazinylcarbonothioyl- and 8-morpholinylcarbonothioyl-derivatives of 1,2-dithiolo[3,4-*c*]quinoline-1-thiones, substituted imino derivative of 1,2-dithiolo[3,4-*c*]pyrrolo[3,2,1-*ij*]quinoline), IC_50_ concentrations (µM) were calculated. The noted high activity of the last compound, which based on in silico predictions is inactive, indicates the novelty of the structure of this molecule in relation to known drugs.

## 2. Results and Discussion

### 2.1. Chemistry

The strategy for the molecular design of DTT-DHQ derivatives was the diversification of the tricyclic structure by introducing various substituents in the aromatic ring and/or to the nitrogen atom of the quinoline fragment, annealing it to five- or six-membered heterocycles, and also the substitution of the exo-sulfur atom. At the same time, by combining a dithioloquinolinethione fragment with some pharmacophores (dioxane, pyrrole) by condensation via the common bond, new chimeric molecules were constructed. New linearly linked hybrid molecules were created by linking the original ligand with various pharmacophore ligands via a metabolizable linker (with piperidine, morpholine, piperazinone, pyrrolidine, isoindole, thiophene, benzathine, etc.). In addition, a hybrid chimeric structure containing both annulated and linearly linked ligands was assembled (Figure 3).

General routes for the synthesis of target compounds and some intermediates are shown in Figure 1. Initial 2,2,4-trimethyl-1,2-dihydroquinolines **1a**–**h** substituted at the aromatic ring or nitrogen atom were obtained according to the known methods [59,60,61]. 8-Heterylcarbonothionyl-DTT-DHQ **2a**–**d** were obtained from the reaction of *N*-alkylhydroquinoline-6-carbaldehydes **1g**,**h** of cyclic secondary amines and excess of elemental sulfur using the previously developed method [62]. Intermediate dithiolo[3,4-*c*]quinoline-1-thiones **2e**–**g** and previously undescribed tetracyclic DTT **2h** were synthesized by sulfurisation of dihydroquinoline **1a**–**d** when refluxed in dimethylformamide with a five-fold excess of sulfur, according to a previously described procedure [63] (Figure 1). Further synthesis of all dithiolothiones **2e**–**h** was carried out according to the methods developed by us earlier [3,4,5,6,7], based on the action of electrophilic reagents on hydrogen atoms in position 6 and/or the secondary amino group of the dihydroquinoline cycle, as well as on the thiocarbonyl group (Figure 1). Therein, previously undescribed DTT derivatives were synthesized. *N*-acyl-[1,2]dithiolo[3,4-*c*]quinoline-1-thiones **2i**–**m** were obtained by the reaction of dithioloquinolines **2f**–**h** with various carbonyl chlorides by reflux in toluene [64,65]. Annelation of the pyrrole-1,2-dione fragment by the Stolle reaction was carried out by the action of oxalyl chloride on dithioloquinoline **2h** by reflux in dry toluene, and pyrrolo[3,2,1-*ij*]quinoline-1,2-dione **2n** was synthesized [66]. Arylamino derivative **2p** was obtained by sequential reactions of alkylation of DTT-DHQ **2e** with methyl iodide and condensation of the resulting iodomethylate **2o** with arylamine **[67]**. Annulated dithiolo[3,4-*c*]pyrrolo[3,2,1-*ij*]quinoline-4,5-dione **2q** was obtained by the acylation of the latter with oxalyl chloride [67].

The structures of the new synthesized compounds **2h**, **2i**–**n**, and **2q** were unambiguously confirmed by ^1^H and ^13^C NMR spectroscopy data and HPLC–HRMS spectrometry. In the ^1^H NMR spectra of all target compounds **2**, proton signals of heme-dimethyl groups were observed in the corresponding fields—at 1.4–2.5 ppm, C(9)-H proton signals shifted in a weak field to 8.6–9.5 ppm. Due to the anisotropic effect of the thioketone group (for compounds **2p**,**q**—arylimino group), the signals of other protons of the quinoline ring appear in the aromatic region of the spectrum [66,67].

The signal of the *NH* proton of the hydroquinoline fragment of compound **2h** was revealed in the characteristic region of 6.0 ppm. Compared to the spectra of the original dithiolothiones **2f**–**h**, in the spectra of compounds **2i**–**n**, no NH proton signal was detected, and in the part of the spectrum corresponding to the aromatic protons of the compound **2n**, one fewer proton was observed. Signals of corresponding *N*-acyl fragments appeared in characteristic regions in the spectra of compounds **2i**–**m** [64,65]. In addition, in the spectra of compounds **2j,k**, signals of two hydrogen atoms of the substituted acetyl fragment were observed in the form of broadened singlets in the characteristic region of 3.8–4.1 ppm [65].

Good-quality spectra of compound **2n** were not obtained due to its low solubility in DMSO. Characteristic signals of the carbon atom of the thiocarbonyl group in the region of 210.7–211.1 ppm were revealed in the ^13^С NMR spectra of compounds **2h** and **2i**–**m**. The signal for the carbon atom of the imino group at 166.6 ppm was detected in the spectrum of the arylimino derivative **2q**. In the mass spectra of compounds **2h**, **2i**–**n**, and **2q**, peaks of protonated molecular ions, consistent with the structure of these compounds, were observed.

### 2.2. Biological Activity Profile Evaluation by PASS

The computer prediction of the biological activity of the simulated compounds was carried out using the PASS Online web resource [36]. PASS Online predicts more than 4300 types of biological activity based on the analysis of a training set containing information on more than 300,000 drug substances and biologically active compounds, with an average accuracy of 95%. The predicted PASS spectrum of biological activity of an organic compound includes pharmacological effects, molecular mechanisms of action, specific toxicity and side effects, metabolism, as well as their effects on undesirable targets, molecular transport, and gene expression. Since PASS allows simultaneous prediction of the interaction of chemical compounds with a large number of biological targets, substances with a pleiotropic effect can be selected based on the results of the prediction. According to the results of the prediction of biological activity using the PASS Online program, almost all dithioloquinolinethiones **2** have multiple activities and, in addition to anti-inflammatory and anti-schistosomal actions, with a probability of more than 50% can possess chemoprotective (58–72%) and antitumor effects (56–66%) and be apoptosis agonists (59–81%) (Table 1). In addition, compounds **2a**–**d**, with a probability of 58–61%, can be dual-specific phosphatase inhibitors. Out of all compounds tested, only for derivative **2q**, containing an arylimino group in the dithiol fragment, the predicted spectrum of activity was very narrow and included only the inhibition of gluconate-2-dehydrogenase (58%).

### 2.3. Biological Evaluation

For the obtained DTT-DHQ derivatives **2a**–**d**, **2h**, **2i**–**n**, and **2q**, primary in vitro screening was carried out for the identification of the leading compounds and determination of their relative inhibitory activity against a number of protein kinases, NPM1-ALK, ALK, EGFR[L858R][T790], cRAF[Y340D][Y341D], JAK2, and JAK3, by ELISA. For the leading compounds, IC_50_ concentrations (µM) were calculated (Table 2). Out of the 12 compounds tested, two compounds, **2k**,**m**, demonstrated very low activity. For most of the chimeric molecules **2h**,**j**,**n**, relative inhibitory activity against the studied protein kinases was moderate (5–56%). In relation to the studied protein kinase hybrid molecules **2a**–**d**,**i**,**l**, they demonstrated inhibitory activity from moderate (12–81%) to high (84–96%) levels, while compounds **2a**–**c** exhibited the maximum non-specific inhibitory effect (IC_50_ 0.25–0.78 µM). The compound **2q**, containing both hybrid and chimeric ligands, showed high activity against kinases JAK3 (99%, IC_50_ 0.46 µM) and cRAF[Y340D][Y341D] (92%, IC_50_ 5.34 µM).

Thus, as a result of testing for the inhibition of protein kinases, substances were identified that exhibit a non-specific inhibitory effect at the nanomolar level against protein kinases JAK3 (IC_50_ 0.36–0.46 μM), NPM1-ALK (IC_50_ 0.25–0.54 μM), and cRAF[Y340D][Y341D] (IC_50_ 0.78 μM) and providing promise for further research.

The highest activity was shown by hybrid derivatives of hydroquinoline dithiolothiones containing a thiocarboxamide group with a morpholine or piperazine residue in the aromatic ring, as well as a hybrid chimeric structure with a pyrroledione fragment fused at the «*i* and *j*» bonds of quinoline and an arylylidene fragment in the dithiol cycle.

## 3. Materials and Methods

### 3.1. Synthesis

NMR ^1^H and ^13^C spectra were registered on a Bruker DRX−500 (500.13 MHz and 125.76 MHz, respectively) spectrometer (Bruker Corporation, Billerica, MA, USA) in DMSO-*d*6, and the internal standard was TMS. Melting points were determined on a Stuart SMP 30 (Cole-Palmer, Staffordshire, UK). To control the reagent and product individually, qualitative analysis of reaction mass was performed by TLC on a Merck TLC Silicagel 60 F254 chromatographic plate (Merck KGaA, Darmstadt, Germany). Eluents included methanol, chloroform, and their mixtures in various rations. The chromatograms were developed by the UV irradiator of chromatographic plates UFS 254/365 Sorbfil (Production company Imid, Krasnodar, Russia) and iodine vapor. Product purity was monitored by high-performance liquid chromatography with high-resolution mass spectrometric electrospray ionization detection (HPLCHRMS–ESI) in combination with UV detection. The analyses were performed on an Agilent 1260 Infinity chromatograph (Agilent Technologies, Santa Clara, CA, USA) and Agilent 6230 TOF LC/MS high-resolution time-of-flight mass detector. The ionization block was double electrospray; the signals were recorded in positive polarity; nebulizer N2 20 psig; desiccant gas N2, 6 mL/min, 325 °C; and mass detection range was 50–2000 daltons. Capillary voltage 4.0 kV, fragmentor +191 V, skimmer +66 V, OctRF 750 V. A Poroshell 120 EC-C18 column (4.6 × 50 mm; 2.7 µm) was used. Gradient elution: acetonitrile/water (0.1% formic acid); flow rate 0.4 mL/min. Software for processing research results: MassHunter Workstation/Data Acquisition V.06.00 (Agilent Technologies, Santa Clara, CA, USA).

Commercially available reagents from Lancaster were also used in the syntheses. The starting compound **1a**–**f** and intermediates **1g**,**h**, **2e**–**g, 2o**,**p** was synthesized according to a published method [59,60,61,63].

Target compounds 8-substituted 4,4,5-trimethyl-4,5-dihydro-1*H*-[1,2]dithiolo[3,4-*c*]quinoline-1-thiones 2a–2d were synthesized according to the procedure developed and described by us earlier [62].

*4,4,5-Trimethyl-8-(morpholin-4-ylcarbonothioyl)-4,5-dihydro-1H-[1,2]dithiolo[3,4-c]quinoline-1-thione* **2a.** Light orange powder, yield 73% (lit. yield 68%), m.p. 106–107 °C (lit. m.p. 105–107 °C [62]); HPLC-HRMS (ESI) calcd for C_18_H_20_N_2_OS_4_ + H^+^, 409.0532; found, 409.0530 (see Appendix A).

*4,4,5-Trimethyl-8-[(4-phenylpiperazin-1-yl)carbonothioyl]-4,5-dihydro-1H-[1,2]dithiolo[3,4-c]quinoline-1-thione***2b.** Light orange powder, yield 82% (lit. yield 80%), m.p. 91–92 °C (lit. m.p. 90–92 °C [62]); HPLC-HRMS (ESI) calcd for C_24_H_25_N_3_S_4_ + H^+^, 484.1005; found, 484.1008 (see Appendix A).

*5-Benzyl-4,4-dimethyl-8-(morpholin-4-ylcarbonothioyl)-4,5-dihydro-1H-[1,2]dithiolo[3,4-c]quinoline-1-thione***2c.** Orange powder, yield 76% (lit. yield 69%), m.p. 111–112 °C (lit. m.p. 110–112 °C [62]); HPLC-HRMS (ESI) calcd for C_24_H_24_N_2_OS_4_ + H^+^, 485.0845; found, 485.0843 (see Appendix A).

*5-Benzyl-4,4-dimethyl-8-(piperidin-1-ylcarbonothioyl)-4,5-dihydro-1H-[1,2]dithiolo[3,4-c]quinoline-1-thione***2d**. Yellow powder, yield 84% (lit. yield 86%), m.p. 98–99 °C (lit. m.p. 97–99 °C [62]); HPLC-HRMS (ESI) calcd for C_25_H_26_N_2_S_4_ + H^+^, 483.1053; found, 483.1054 (see Appendix A).

Procedure for the synthesis of 4,4-dimethyl-4,5,8,9-tetrahydro-1*H*-[1,4]dioxino[2,3-*g*][1,2]dithiolo[3,4-*c*]quinoline-1-thione **2h**: a mixture of quinolines **1d** (4.62 g, 20 mmol) and elemental sulfur (3.2 g, 100 mmol) in DMF (20 mL) was refluxed for 15 h. The reaction mixture was poured into water, and the precipitate was recrystallized from toluene to furnish the desired product **2h**. Brown-red powder (yield 5.23 g, 81%), m.p. 238–239 °C; ^1^H NMR (DMSO-d_6_, 500 MHz) δ ppm: 1.48 (br s, 6H, (CH_3_)_2_), 4.16 (q, *J* = 4.8 Hz, 2H, CH_2_), 4.23 (q, *J* = 4.8 Hz, 2H, CH_2_), 6.02 (s, 1H, NH), 6.32 (s, 1H, H-6 quinoline), 8.84 (s, 1H, H-9 quinoline); ^13^C NMR (DMSO-d_6_, 125 MHz) δ ppm: 27.2, 40.1, 56.0, 63.8, 64.6, 102.4, 110.9, 111.6, 134.0, 134.1, 138.3, 144.5, 175.4, 210.8; HPLC-HRMS (ESI) calcd for C_14_H_13_NO_2_S_3_ + H^+^, 324.0182; found, 324.0180 (see Appendix A).

General procedure for synthesis of substituted N-acyl-[1,2]dithiolo[3,4-c]quinoline-1-thione **2i**–**m**: to a solution of the starting compounds **2f**–**h** (5 mmol) in dry toluene (10 mL), a solution of corresponding acylchloride (5.5 mmol) in toluene (10 mL) was added dropwise under cooling. The reaction mixture was refluxed for 8–10 h while the reaction progress was monitored by TLC. Toluene was distilled off under reduced pressure, and the precipitate was filtered and recrystallized from toluene to furnish the desired products **2i**–**m**.

*8-Methoxy-4,4-dimethyl-5-(2-thienylcarbonyl)-4,5-dihydro-1H-[1,2]dithiolo[3,4-c]quinoline-1-thione* (**2i**). Orange powder (yield 1.76g, 87%), m.p. 158–159 °C; ^1^H NMR (DMSO-d_6_, 500 MHz) δ ppm: 1.81 (br s, 6H, (CH_3_)_2_), 3.73 (s, 3H, CH_3_O), 6.78 (d, *J* = 8.7 Hz, 2H, H-6, H-7 quinoline); 6.95 (t, *J* = 4.2 Hz, 1H, H-5(3) Ar), 7.08 (d, *J* = 4.2 Hz 1H, H-3(5) Ar), 7.77 (d, *J* = 7.96 Hz, 1H, H-Ar), 7.72 (d, *J* = 4.2 Hz, 1H, H-4 Ar), 8.88 (s, 1H, H-9 quinoline); ^13^C NMR (DMSO-d_6_, 125 MHz) δ ppm: 25.1, 40.1, 55.3, 61.7, 108.1, 114.3, 124.2, 126.3, 127.6, 130.5, 133.0, 133.5, 133.9, 140.0, 155.6, 163.2, 179.6, 211.5; HPLC-HRMS (ESI) calcd for C_18_H_15_NO_2_S_4_ + H^+^, 406.0059; found, 406.0060 (see Appendix A).

*1-[2-(8-Methoxy-4,4-dimethyl-1-thioxo-1,4-dihydro-5H-[1,2]dithiolo[3,4-c]quinolin-5-yl)-2-oxoethyl]pyrrolidine-2,5-dione* (**2j**). Orange powder (yield 1.54 g, 71%), m.p. = 201–202 °C; ^1^H NMR (DMSO-d_6_, 500 MHz) δ ppm: 1.27 (br s, 3H, (CH_3_)_2_), 2.15 (br s, 3H, (CH_3_)_2_), 2.51 (s, 2H, CH_2_), 3.57 (s, 2H, CH_2_), 3.81 (s, 3H, CH_3_O), 4.07 (br s, 2H, CH_2_CO), 7.03 (d, *J* = 8.7 Hz, 1H, H-6(7) quinoline), 7.58 (d, *J* = 8.7 Hz, 1H, H-7(6) quinoline), 8.77 (s, 1H, H-9 quinoline); ^13^C NMR (DMSO-d_6_, 125 MHz) δ ppm: 26.6, 27.1, 27.9, 42.4, 55.5, 61.6, 66.4, 108.7, 114.5, 126.9, 127.3, 127.8, 134.1, 157.3, 167.6, 176.7, 181.6, 211.1; HPLC-HRMS (ESI) calcd for C_19_H_18_N_2_O_4_S_3_ + H^+^, 435.0503; found, 435.0503 (see Appendix A).

*2-[2-(8-Methoxy-4,4-dimethyl-1-thioxo-1,4-dihydro-5H-[1,2]dithiolo[3,4-c]quinolin-5-yl)-2-oxoethyl]hexahydro-1H-isoindole-1,3(2H)-dione* (**2k**). Orange powder (yield 1.78 g, 73%), m.p. = 205–206 °C; ^1^H NMR (DMSO-d_6_, 500 MHz) δ ppm: 1.27–2.50 (m, 14H, 8H, 4 CH_2_ + 6H, (CH_3_)_2_), 2.84 (s, 2H, CH), 3.81 (s, 3H, CH_3_O), 4.06 (br s, 2H, CH_2_CO), 7.04 (d, *J* = 8.5 Hz, 1H, H-6(7) quinoline), 7.59 (d, *J* = 8.5 Hz, 1H, H-7(6) quinoline), 8.79 (s, 1H, H-9 quinoline); ^13^C NMR (DMSO-d_6_, 125 MHz) δ ppm: 21.2, 23.0, 24.7, 25.9, 26.1, 26.4, 39.0, 41.9, 55.5, 61.6, 108.8, 114.4, 127.1, 127.5, 127.8, 134.1, 157.3, 167.8, 178.7, 181.6, 211.0; HPLC-HRMS (ESI) calcd for C_23_H_24_N_2_O_4_S_3_ + H^+^, 489.0972; found, 489.0970 (see Appendix A).

*5-[(3-Chloro-1-benzothien-2-yl)carbonyl]-7-methoxy-4,4-dimethyl-4,5-dihydro-1H-[1,2]dithiolo[3,4-c]quinoline-1-thione* (**2l**). Orange powder (yield 2.12 g, 87%), m.p. = 218–219 °C; ^1^H NMR (DMSO-d_6_, 500 MHz) δ ppm: 1.88 (br s, 6H, (CH_3_)_2_), 3.50 (s, 3H, CH_3_O), 6.48 (s, 1H, H-6 quinoline), 6.77 (d, *J* = 8.8 Hz, 1H, H-8 quinoline), 7.51 (t, *J* = 7.9 Hz, 1H, H-5(6) Ar), 7.54 (d, *J* = 7.9 Hz, 1H, H-6(5) Ar), 7.73 (d, *J* = 7.9 Hz, 1H, H-4(7) Ar), 8.02 (d, *J* = 7.9 Hz, 1H, H-7(4) Ar), 9.11 (d, *J* = 8.8 Hz, 1H, H-9 quinoline); ^13^C NMR (DMSO-d_6_, 125 MHz) δ ppm: 25.5, 55.2, 62.3, 110.2, 110.3, 117.5, 121.7, 122.8, 123.5, 123.9, 126.1, 128.0, 133.5, 134.6, 135.2, 137.3, 137.4, 158.8, 158.9, 161.9, 175.8, 211.1; HPLC-HRMS (ESI) calcd for C_22_H_16_ClNO_2_S_4_ + H^+^, 489.9826; found, 489.9823 (see Appendix A).

*5-(2-Fluorobenzoyl)-4,4-dimethyl-4,5,8,9-tetrahydro-1H-[1,4]dioxino[2,3-g][1,2]dithiolo[3,4-c]quinoline-1-thione* (**2m**). Orange powder (yield 1.54 g, 69%), m.p. = 234–235 °C; ^1^H NMR (DMSO-d_6_, 500 MHz) δ ppm: 1.86 (br s, 6H, (CH_3_)_2_), 2.23 (s, 3H, CH_3_), 4.14 (d, *J* = 10.05 Hz, 4H, 2 CH_2_O), 6.34 (s, 1H, H-6 quinoline), 7.06 (t, *J* = 8.7 Hz, 1H, H-4(5) Ar), 7.17 (t, *J* = 8.7 Hz, 1H, H-5(4) Ar), 7.42 (d, *J* = 8.7 Hz, 2H, H-3 + H-6 Ar), 8.67 (s, 1H, H-9 quinoline); ^13^C NMR (DMSO-d_6_, 125 MHz) δ ppm: 25.8, 61.9, 64.0, 111.1, 114.6, 115.8, 118.4, 124.7, 125.9, 129.81, 130.5, 132.7, 134.1, 140.2, 142.7, 157.5, 159.2, 165.5, 177.7, 210.9; HPLC-HRMS (ESI) calcd for C_21_H_16_FNO_3_S_3_ + H^+^, 446.0350; found, 446.0353 (see Appendix A).

General procedure for synthesis of substituted 1,2-dithiolo[3,4-c]pyrrolo[3,2,1-ij]quinolinediones **2n**,**q**: to a solution of the starting compounds **2h** (or **2p**) (5 mmol) in dry toluene (20 mL), an oxalyl chloride (0.7g, 0.47 mL, 5.5 mmol) was added. The reaction mixture was refluxed for 1.5–2 h while the reaction progress was monitored by TLC. Toluene was distilled off under reduced pressure, the precipitate was filtered, washed with ethanol, and the resulting target products **2n** (or **2q**) did not require recrystallization.

*8,8-Dimethyl-11-thioxo-2,3,8,11-tetrahydro[1,4]dioxino[2,3-g][1,2]dithiolo[3,4-c]pyrrolo[3,2,1-ij]quinoline-5,6-dione* (**2n**). Dark brown powder (yield 1.58 g, 84%), m.p. > 260 °C; ^1^H NMR (DMSO-d_6_, 500 MHz) δ ppm: 2.05 (s, 6H, (CH_3_)_2_), 4.28 (q, *J* = 3.5 Hz*,* 2H, CH_2_O), 4.43 (q, *J* = 3.5 Hz*,* 2H, CH_2_O), 9.35 (s, 1H, H-9 quinoline); HPLC-HRMS (ESI) calcd for C_16_H_11_NO_4_S_3_ + H^+^, 377.9924; found, 377.9925 (see Appendix A).

*10-[(3-Methoxyphenyl)imino]-7,7-dimethyl-7,10-dihydro[1,2]dithiolo[3,4-c]pyrrolo[3,2,1-ij]quinoline-4,5-dione* (**2q**). Brown-red powder (yield 1.70 g, 87%), m.p. = 195–196 °C; ^1^H NMR (DMSO-d_6_, 500 MHz) δ ppm: 2.01 (br s, 6H, (CH_3_)_2_), 3.33 (s, 3H, CH_3_O), 6.64 (d, *J* = 7.8 Hz, 1H, H-8 quinoline),6.48 (s, 1H, H-6 quinoline), 6.77 (d, *J* = 8.8 Hz, 1H, H-8 quinoline); ^13^C NMR (DMSO-d_6_, 125 MHz) δ ppm: 27.5, 55.3, 60.5, 104.9, 111.1, 111.3, 115.1, 115.6, 119.6, 122.9, 123.6, 130.6, 131.1, 146.7, 152.9, 157.8, 160.8, 163.0, 166.6, 182.0; HPLC-HRMS (ESI) calcd for C_21_H_16_N_2_O_3_S_2_ + H^+^, 409.0676; found, 409.0679 (see Appendix A).

### 3.2. Virtual Screening

PASS Online allows registered users to receive a prediction of the spectrum of biological activity based on the structural formula of a chemical compound via the Internet (http://www.way2drug.com/passonline, accessed on 1 January 2022) [36]. The forecast results are presented in Table 1 as a list of names of probable activities for the indicated compounds with calculated estimates of the probabilities of presence (Pa) or absence (Pi) of activity, which have values from 0 to 1. Since the probabilities are calculated independently on subsamples of active or inactive compounds, respectively, their sum is not is equal to one. The values of Pa and Pi are interpreted as estimates of the measure of whether a substance belongs to the classes of active and inactive compounds, respectively. The greater the value of Pa for a particular activity and the smaller Pi, the higher the chance of detecting this activity in the experiment. If Pa < 0.5, but Pa > Pi, the probability of detecting this type of activity is low, but if the activity is found, then there is a more than 50% chance that the Pa and Pi structure is original [36].

### 3.3. Biological Evaluation

Study of inhibitory activity. The kinase activity was determined by enzyme-linked immunosorbent assay (ELISA) in polypropylene plates (Costar, 3363) in reaction buffer (20 mM HEPES, pH 7.5, 15 mM MgCl2, 2 mM DTT, 0.2 mM Na_3_VO_4_, 0.005% Triton X-100) for 60 min at 30 °C with vigorous stirring. The final concentration of the reaction components: 0.05 μg mL^–1^ of the corresponding kinase, 5 nM Histon H3 biotinylated substrate (1–21) (Anaspec, 61702), 150 μM ATP (Sigma, A6419), 10 μM test compound, 5% DMSO. The enzymatic reaction was stopped with a buffer containing 20 mM HEPES (Sigma, H4034), pH 7.5, and 150 mM EDTA (Sigma, E5513). Further, to detect the phosphorylated substrate, the reaction mixture was transferred to pre-prepared plates (Nunc, 468667) coated with neutravidin (1 ng per well; Pierce, 31000) and treated with bovine serum albumin (BSA) to block non-specific binding sites. Incubation was carried out for 1 h at room temperature. After washing the plates three times with phosphate buffer saline (PBS, pH 7.4) with Tween-20, they were incubated sequentially with anti-phospho-Histon H3 antibodies (0.3 ng μL^–1^; Millipore, 04-746) and with specific antibodies conjugated with an enzyme-label (peroxidase) Anti-rabbit IgG, HRP-linked Antibody (titer 1/5000; Cell Signaling, 7074). After completion of each incubation step (60 min at room temperature (~20 °C) and continuous stirring), the plates were washed three times from unbound antibody molecules with a PBS solution with Tween-20 and 100 μL of substrate (3.3′,5.5′-tetramethylbenzidine (TMB) dihydrochloride, Sigma, T8768) prepared according to the manufacturer’s instructions. Before measuring the optical density, the reaction was stopped using 0.5 M Н_2_SO_4_. The optical density of the solution was determined at λ = 450 nm using a tablet spectrophotometer (TECAN Safire). The data obtained were processed and imported into the HTSCalc program.

For compounds **2a**–**c**,**q**, the IC_50_ value (μmol L^–1^) was calculated from ten concentration points (in triplicate dilution) obtained twice. The data were analyzed using GraphPad Prism software (v. 3.1).

## 4. Conclusions

Thus, compounds with pleiotropic activity, including chemoprotective and antitumor activity, were found among previously synthesized and obtained new hybrid and chimeric derivatives of 1,2-dithiolo[3,4-*c*]quinoline-1-thione using the PASS Online software. Experimental testing of the activity towards a number of protein kinases, NPM1-ALK, ALK, EGFR[L858R][T790], cRAF[Y340D][Y341D], JAK2, and JAK3, confirmed the calculated predictions. The most active non-selective kinase inhibitors were 8-phenylpiperazinylcarbonothioyl and 8-morpholinylcarbonothioyl derivatives of dithiolthionodihydroquinolines. In addition, high activity towards protein kinases cRAF[Y340D][Y341D] and JAK3 for the substituted imino-derivative of 1,2-dithiolo[3,4-c]pyrrolo[3,2,1-ij]quinoline, which according to in silico predictions was inactive, was determined. This finding indicates the novelty of the structure of the molecule in relation to known drugs. Considering the need to develop new non-selective kinase inhibitors, derivatives of dithioloquinolinethione and dithioloquinolineimine require further study as promising preferable structures for the treatment of multifactorial diseases such as cancer, diabetes, and inflammatory processes of various etiologies.

## Data Availability

Not applicable.

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
