# Peer review of "Synthesis of 4,5-Dihydro-1H-[1,2]dithiolo[3,4-c]quinoline-1-thione Derivatives and Their Application as Protein Kinase Inhibitors"

_molecules, 2022, doi:10.3390/molecules27134033_

Round 1

Reviewer 1 Report

Medvedeva and Shikhaliev report the synthesis of some 4,5-Dihydro-1H-[1,2]dithiolo[3,4-c]quinoline-1-thiones and their propeinkinase inhibitory activity. This manuscript is well written and presented. This reviewer recommends publication in Molecules in the present form.

Author Response

Thank you for recommending our article for publication in the journal

Reviewer 2 Report

in the present study, the authors reported the synthesis and characterization of a set of 4,5-Dihydro-1H-[1,2]dithiolo[3,4-c]quinoline-1-thi-2 ones analogues. The authors have investigated the inhibitory potency of synthesized compounds toward the propein kinase activity. The results showed that among the synthesized compounds, two compounds exhibited potent inhibitory activity toward different kinases. Overall, this is interesting. However, the manuscript is really poorly written and the results are presented in confusing way that make it hardly to understand. Further, the manuscript requires extensive editing for academic style and for grammatical and typo mistakes. Accordingly, I would not suggest the publication of this manuscript in the present form. The following are the major concerns:

- the synthetic scheme is really poorly presented, the authors must change the format and present the different R, R`, R`` under the compounds not as legend. Further the conditions of reaction should be noted as a,b,c, d not C1,D2, and so on

- Indeed, it is not clear how many novel derivatives have been synthesized. In the material and methods the authors mentioned The starting compound 1a-f, intermediates 1g,h, 2e-h, 2o,p and target compounds 2a-d was synthesized according to a published method [59-67]'' which means that only 2i-2n are the new compounds??

- Toward characterization of synthesized compounds, the authors must detail all analytical data for ALL compounds synthesized in the material and methods section.

- The NMR spectra supported are not correctly calibrated. The no of hydrogens should be corrected otherwise it is not possible to judge the purity and identity of synthesized compounds

- Please correct the structures provided in the spectra in the supplementary information

- Regarding the biological data, the authors must include a positive control drug in order to prove the activity presented for their compounds. Otherwise, the presented data is not fully proved.

Minor concerns:

- the abstract is poorly written. Please re-write it and present the IC50 not as a range e.g.(IC50 0.25-0.78 μM), But as specific activity for the compound e.g., 8-morpholinylcarbonothioyl-derivatives of 1,2-dithiolo[3,4-21 c]quinoline-1-thiones (IC50 =0.25uM).

- in the introduction figures, please avoid colored atoms and add better resolution figures for Fig1-3

- avoid word interaction when you describe a chemical reaction, use instead reaction (e.g., line 151).

- inline 160, At the same time, previously unknown derivatives of DTT were synthesized, what does it mean?

- in table 1: what the underline means in the table?

Author Response

Thank you for reviewing our manuscript, for your comments and suggestions. Corrections, according to your comments, have been made to the article and this has improved it. It remains to fix the bad translation into English, which will take some time. The article will be posted after the English is improved.

Comments and Suggestions for Authors

in the present study, the authors reported the synthesis and characterization of a set of 4,5-Dihydro-1H-[1,2]dithiolo[3,4-c]quinoline-1-thi-2 ones analogues. The authors have investigated the inhibitory potency of synthesized compounds toward the propein kinase activity. The results showed that among the synthesized compounds, two compounds exhibited potent inhibitory activity toward different kinases. Overall, this is interesting. However, the manuscript is really poorly written and the results are presented in confusing way that make it hardly to understand. Further, the manuscript requires extensive editing for academic style and for grammatical and typo mistakes. Accordingly, I would not suggest the publication of this manuscript in the present form. The following are the major concerns:

- the synthetic scheme is really poorly presented, the authors must change the format and present the different R, R`, R`` under the compounds not as legend. Further the conditions of reaction should be noted as a,b,c, d not C1,D2, and so on

Thanks for the advice, the synthetic scheme has been corrected.

- Indeed, it is not clear how many novel derivatives have been synthesized. In the material and methods the authors mentioned The starting compound 1a-f, intermediates 1g,h, 2e-h, 2o,p and target compounds 2a-d was synthesized according to a published method [59-67]'' which means that only 2i-2n are the new compounds??

We have studied the antikinase activity of 12 compounds synthesized by us. Of these compounds, 8 (2h-n, q) were synthesized for the first time in this work; therefore, the procedures for their synthesis and all their spectral characteristics are given. The procedure for the synthesis of 4 compounds (2a-d) was developed by the author Shikhaliev earlier, their full characteristics were described (62. Manahelohe, G.M.; Shikhaliev, K.S.; Potapov, A.Y. Synthesis of 1H-1,2-dithiol-1-thiones and thioamides containing hydro-quinoline group Eur Chem Bull 2015 4 350-355 https://doi.org/10.17628/ECB.2015.4.350). In order not to repeat the already published information on these compounds, in this work, in the experimental part, we do not describe the procedure by which compounds 2a-d were synthesized, but refer to the previous work [62]. We also do not describe the procedures for the synthesis of the previously described starting materials and intermediates, but refer to previous publications [59-67].

- Toward characterization of synthesized compounds, the authors must detail all analytical data for ALL compounds synthesized in the material and methods section.

In the Materials and Methods section, we have indicated all physicochemical and spectral characteristics for all newly synthesized compounds 2h-n, q. For compounds 2a-d, the complete characteristics and synthesis of which were described by the author earlier, only melting points and yields are given in comparison with those published [62]. The purity of these compounds is confirmed by the reported data of analysis using the HPLC-MS method. This approach to describing the characteristics of compounds was made by us in an article published this year:Тashchilova, A.; Podoplelova, N.; Sulimov, A.; Kutov, D.; Ilin, I.; Panteleev, M.; Shikhaliev, K.; Medvedeva, S.; Novichikhina, N.; Potapov, A.; et al. New Blood Coagulation Factor XIIa Inhibitors: Molecular Modeling, Synthesis, and Experimental Confirmation. Molecules 2022, 27, 1234. https://doi.org/10.3390/molecules27041234

- The NMR spectra supported are not correctly calibrated. The no of hydrogens should be corrected otherwise it is not possible to judge the purity and identity of synthesized compounds

Thank you, corrected

- Please correct the structures provided in the spectra in the supplementary information

The structures are shown correctly, if they interfere with the spectrum, then we can remove them.

- Regarding the biological data, the authors must include a positive control drug in order to prove the activity presented for their compounds. Otherwise, the presented data is not fully proved.

Thank you, we have added information on the control drug - sorafenib

Minor concerns:

- the abstract is poorly written. Please re-write it and present the IC50 not as a range e.g.(IC50 0.25-0.78 μM), But as specific activity for the compound e.g., 8-morpholinylcarbonothioyl-derivatives of 1,2-dithiolo[3,4-21 c]quinoline-1-thiones (IC50 =0.25uM).

Thank you, the annotation has been corrected according to your recommendations.

- in the introduction figures, please avoid colored atoms and add better resolution figures for Fig1-3

Thank you, corrected

- avoid word interaction when you describe a chemical reaction, use instead reaction (e.g., line 151).

Thank you, corrected

- inline 160, At the same time, previously unknown derivatives of DTT were synthesized, what does it mean?

This means that the DTT derivatives were newly synthesized. Thank you, the phrase was poorly translated, it has been corrected.

- in table 1: what the underline means in the table?

Underlining separated the values of Pa from Pi. Thanks, separated with "/".

Reviewer 3 Report

The article “Synthesis of 4,5-Dihydro-1H-[1,2]dithiolo[3,4-c]quinoline-1-thi-ones Derivatives and Their Application as Propeinkinase (?) Inhibitors” highlights the importance of 3H-1,2-dithiol-3-thione (DTT) derivatives in medicinal chemistry, focusing on a new series of dithioloquinolinethiones. The manuscript describes in depth synthesis, in silico activity and in vitro screening. In particular, the inhibitory effect was evaluated against a large number of protein kinases NPM1-ALK, ALK, EGFR[L858R][T790], cRAF[Y340D][Y341D], JAK2, JAK3, toward which several compounds showed sub-micromolar activity. The articles is well-organized, properly describes results and material and methods. Hence, it is suitable for publications after the following minor revisions:

-          There is a typo in the title.

-          Please change Figures 1, 2 and 3 with a high-quality images.

-          English should be revised in some points. For example, rephrase lines 63-64.

-          At the end of line 90, authors must report some studies that describe the correlation between “mutation processes and/or activation of enzymes of the protein kinase family” in cancer, diabetes and inflammatory processes. Herein, some suggestions: Eur J Med Chem, 2022, 16; 235:114292. doi: 10.1016/j.ejmech.2022.114292; ACS Med. Chem. Lett. 2022, 13, 3, 358 - 364 https://doi.org/10.1021/acsmedchemlett.1c00600; Eur. J. Cancer 138S2 (2020) S1–S62 https://doi.org/10.1016/S0959-8049(20)31181-3.

-          In the synthetic scheme, each method should have a different letter. Avoid A, B1, B2, C1, C2 and use a,b,c,d .. or i,ii,iii,iv, etc.

-          The description of NMR spectra (lines 177-199) should be inserted in the supplementary table, not in the discussion.

-          The IC50 values of compounds 2a-c and 2q against CRAF [Y340D] [Y341D], JAK3 and NPM1- ALK should be fully reported in table 2. The data against CRAF [Y340D] [Y341D] for 2a,2b and the data against NPM1- ALK for 2c, 2q are missing.

Author Response

Thank you for reviewing our manuscript, for your comments and suggestions. Corrections, according to your comments, have been made to the article and this has improved it. It remains to fix the bad translation into English, which will take some time. The article will be posted after the English is improved.

The article “Synthesis of 4,5-Dihydro-1H-[1,2]dithiolo[3,4-c]quinoline-1-thi-ones Derivatives and Their Application as Propeinkinase (?) Inhibitors” highlights the importance of 3H-1,2-dithiol-3-thione (DTT) derivatives in medicinal chemistry, focusing on a new series of dithioloquinolinethiones. The manuscript describes in depth synthesis, in silico activity and in vitro screening. In particular, the inhibitory effect was evaluated against a large number of protein kinases NPM1-ALK, ALK, EGFR[L858R][T790], cRAF[Y340D][Y341D], JAK2, JAK3, toward which several compounds showed sub-micromolar activity. The articles is well-organized, properly describes results and material and methods. Hence, it is suitable for publications after the following minor revisions:

-          There is a typo in the title.

-          Please change Figures 1, 2 and 3 with a high-quality images.

Thank you, corrected

-          English should be revised in some points. For example, rephrase lines 63-64.

Thank you, corrected

-          At the end of line 90, authors must report some studies that describe the correlation between “mutation processes and/or activation of enzymes of the protein kinase family” in cancer, diabetes and inflammatory processes. Herein, some suggestions: Eur J Med Chem, 2022, 16; 235:114292. doi: 10.1016/j.ejmech.2022.114292; ACS Med. Chem. Lett. 2022, 13, 3, 358 - 364 https://doi.org/10.1021/acsmedchemlett.1c00600; Eur. J. Cancer 138S2 (2020) S1–S62 https://doi.org/10.1016/S0959-8049(20)31181-3.

Thank you, 2 links have been added

-          In the synthetic scheme, each method should have a different letter. Avoid A, B1, B2, C1, C2 and use a,b,c,d .. or i,ii,iii,iv, etc.

Thank you, corrected

-          The description of NMR spectra (lines 177-199) should be inserted in the supplementary table, not in the discussion.

In the discussion of the results for newly synthesized substances, it is necessary to provide information confirming the correctness of the structure attributed to them, and this is done on the basis of spectral data. Not only we think so, for example, the authors of the article (El-Sayed, W.A.; Alminderej, F.M.; Mounier, M.M.; Nossier, E.S.; Saleh, S.M.; Kassem, A.F. Novel 1,2,3-Triazole-Coumarin Hybrid Glycosides and Their Tetrazolyl analogues: Design, Anticancer Evaluation and Molecular Docking Targeting EGFR, VEGFR-2 and CDK-2 (Molecules 2022, 27, 2047. https://doi.org/10.3390/molecules27072047) also describe the spectra in the discussion.

-          The IC50 values of compounds 2a-c and 2q against CRAF [Y340D] [Y341D], JAK3 and NPM1- ALK should be fully reported in table 2. The data against CRAF [Y340D] [Y341D] for 2a,2b and the data against NPM1- ALK for 2c, 2q are missing.

IC50 values were determined only for compounds that showed the maximum percentage of inhibition (>85%) in relation to CRAF [Y340D] [Y341D], JAK3 and NPM1-ALK, therefore, data on NPM1-ALK for 2c, 2q are not available. Data on CRAF [Y340D] [Y341D] for compound 2a added. Compound 2b was inactive against CRAF [Y340D] [Y341D].

Round 2

Reviewer 2 Report

the authors have adequately addressed all concerns that have been raised.